# Herpes simplex type 2 virus deleted in glycoprotein D protects against vaginal, skin and neural disease

Christopher Petro[1,2,3†], Pablo A González[1,2,4†], Natalia Cheshenko[1,3], Thomas Jandl[1,3], Nazanin Khajoueinejad[3], Angèle Bénard[1], Mayami Sengupta[1,2], Betsy C Herold[1,3]*, William R Jacobs Jr[1,2]*

[1]Department of Microbiology and Immunology, Albert Einstein College of Medicine, New York, United States; [2]Howard Hughes Medical Institute, Albert Einstein College of Medicine, New York, United States; [3]Department of Pediatrics, Albert Einstein College of Medicine, New York, United States; [4]Millennium Institute on Immunology and Immunotherapy, Facultad de Ciencias Biológicas, Pontificia Universidad Católica de Chile, Santiago, Chile

**Abstract** Subunit vaccines comprised of glycoprotein D (gD-2) failed to prevent HSV-2 highlighting need for novel strategies. To test the hypothesis that deletion of gD-2 unmasks protective antigens, we evaluated the efficacy and safety of an HSV-2 virus deleted in gD-2 and complemented allowing a single round of replication on cells expressing HSV-1 gD ($\Delta gD^{-/+gD-1}$). Subcutaneous immunization of C57BL/6 or BALB/c mice with $\Delta gD^{-/+gD1}$ provided 100% protection against lethal intravaginal or skin challenges and prevented latency. $\Delta gD^{-/+gD1}$ elicited no disease in SCID mice, whereas 1000-fold lower doses of wild-type virus were lethal. HSV-specific antibodies were detected in serum (titer 1:800,000) following immunization and in vaginal washes after intravaginal challenge. The antibodies elicited cell-mediated cytotoxicity, but little neutralizing activity. Passive transfer of immune serum completely protected wild-type, but not Fcγ-receptor or neonatal Fc-receptor knock-out mice. These studies demonstrate that non-neutralizing Fc-mediated humoral responses confer protection and support advancement of this attenuated vaccine.

*For correspondence: betsy.herold@einstein.yu.edu (BCH); jacobsw@hhmi.org (WRJ)

†These authors contributed equally to this work

Competing interests: The authors declare that no competing interests exist.

## Introduction

Herpes simplex viruses (HSV) are major global health problems. HSV serotype 1 (HSV-1) is associated primarily with oral mucocutaneous disease, sporadic encephalitis, and is a leading cause of corneal blindness worldwide, whereas HSV serotype 2 (HSV-2) is the leading cause of genital ulcerative disease and a major cofactor in fueling the HIV epidemic (*Gray et al., 2001*; *Wald and Link, 2002*; *Freeman et al., 2006*). The World Health Organization estimated that over 500 million people are infected with HSV-2 worldwide with approximately 20 million new cases annually (*Looker et al., 2008*). The extremely high prevalence of HSV-2 in sub-Saharan Africa (~70%) (*Looker et al., 2008*) may contribute more to the spread of HIV-1 than number of sex partners or other sexually transmitted infections (*Freeman et al., 2006*; *Chen et al., 2007*). Moreover, as HSV establishes latency in neurons with frequent subclinical or clinical reactivations, there is a lifelong impact of infection. Notably, HSV-1 has emerged as a predominant cause of genital disease in the developed world (*Roberts et al., 2003*; *Xu et al., 2006*; *Horowitz et al., 2011*; *Bernstein et al., 2013*). These epidemiological findings highlight the urgency to develop a safe and effective vaccine that protects against both serotypes.

Subunit vaccines comprised of the HSV-2 viral envelope glycoprotein D (gD-2) alone or combined with other envelope glycoproteins and with different adjuvants have predominated the HSV vaccine

**eLife digest** Herpes simplex virus 2 (or HSV-2) infects millions of people worldwide and is the leading cause of genital diseases. The virus initially infects skin cells, but then spreads to nerve cells where it persists for life. Often, the virus remains in a dormant state for long periods of time and does not cause any symptoms. However, HSV-2 can periodically re-activate, leading to repeated infections; this can be life-threatening in patients who suffer from a weak immune system. There is no cure for Herpes simplex virus infection, and there are currently no vaccines that would prevent the virus from infecting humans.

HSV-2 contains a protein on its surface known as 'glycoprotein D' which it needs to enter host cells. The interaction between glycoprotein D and the host is also essential for cell-to-cell spread of the virus. Vaccines that contain glycoprotein D trigger the production of antibodies that bind to this viral protein. These vaccines have been tested in several large clinical trials, but the results have so far been disappointing. As such, new vaccines that provide effective protection against HSV-2 are urgently needed.

Live attenuated vaccines are commonly used to prevent diseases such as measles mumps and chicken pox or shingles. These vaccines contain a harmless or weakened version of the disease-causing virus. Petro, González et al. have now developed a new potential vaccine that contains live attenuated HSV-2, which completely lacks glycoprotein D and thus cannot spread from cell-to-cell. When this weakened virus was administered to mice that have a poor immune system, the mice remained healthy. On the other hand, when Petro, González et al. treated similar mice with the wild-type HSV-2 virus instead, many mice died within a few days.

Petro, González et al. then went on to show that mice that had been treated with the weakened virus as a vaccine were completely protected from a later infection with wild-type HSV-2 and did not develop any symptoms of the disease. Furthermore, no virus was detected in the nerve cells of these mice—which is where the virus would normally persist in its dormant state. Finally, Petro, González et al. showed that blood serum from immunized mice could be used to completely protect other mice from exposure to wild-type virus. These results demonstrate that a live attenuated HSV-2 virus that lacks glycoprotein D (the main component of other failed vaccines) elicits a different type of immune response and is a safe and effective vaccine in mouse models of virus infection. With further work, these findings may eventually lead to a preventative treatment to combat HSV-2 infections in humans.

field for nearly 20 years (*Mertz et al., 1990*; *Corey et al., 1999*; *Stanberry et al., 2002*; *Bernstein et al., 2005*; *Belshe et al., 2012*; *Leroux-Roels et al., 2013*). The rationale for gD-2 subunit vaccines included their safety and ability to elicit neutralizing antibody (Ab) responses in vitro. However, the clinical trial outcomes have been disappointing. For example, in the most recent efficacy trial, an HSV-2 gD adjuvanted subunit vaccine provided no protection against HSV-2 genital disease, but surprisingly provided 35% cross-protection against HSV-1 infection and 58% cross-protection against HSV-1 disease (*Belshe et al., 2012*). Neutralizing Abs (1:422 against HSV-1 and 1:97 against HSV-2) were detected in the serum, but were not measured in mucosal sites (*Awasthi et al., 2014*; *Belshe et al., 2014*). The gD-2 subunit vaccine also elicited CD4+, but not CD8+ T cell responses; however, the cellular responses did not correlate with HSV-1 protection (*Belshe et al., 2014*). These findings suggest the need for more complex immunogens to elicit protective immune responses against both serotypes.

The efficacy of the live attenuated varicella zoster virus (VZV) (a related herpes virus) vaccine suggests that a similar approach for HSV may prove effective. However, preclinical studies with attenuated HSV vaccine candidates have yielded variable but incomplete protection against primary infection and latency (*Boursnell et al., 1997*; *Spector et al., 1998*; *Hoshino et al., 2008*; *Shin and Iwasaki, 2012*). For example, *Shin* and *Iwasaki* found that a 'prime-pull' strategy, in which mice were first primed subcutaneously with an attenuated HSV deleted in thymidine kinase and then treated intravaginally with chemokines to recruit protective CD8+ T cells into the genital tract, was required for protection (*Shin and Iwasaki, 2012*). We adopted a different attenuated vaccine strategy based on an HSV-2 virus genetically deleted for the gene that encodes gD-2 ($U_S6$) and complemented the

virus to allow a first round of replication by growing it on Vero cells encoding HSV-1 $U_S6$ (gD-1, VD60 cells) (*Cheshenko et al., 2013*). The HSV-1 gD complemented HSV-2 mutant virus (designated HSV-2 $\Delta gD^{-/+gD-1}$) replicates in VD60 cells to high titers, but a single passage through non-complementing cells yields non-infectious progeny, reflecting the requirement for gD in entry and cell-to-cell spread. We evaluated this virus for safety, immunogenicity, and vaccine efficacy against intravaginal challenge and in a skin scarification model in different strains of mice. Subcutaneous prime and boost of mice with HSV-2 $\Delta gD^{-/+gD-1}$ protected against lethal challenge in both models and prevented the establishment of latency. Adoptive transfer experiments showed that protection was mediated by antibodies (Abs) and required both the Fcγ receptor and neonatal Fc receptor (FcRn).

## Results

### Immunization with HSV-2 $\Delta gD^{-/+gD-1}$ is safe and protects mice from intravaginal lethal challenge

We reasoned that deletion of gD-2, an envelope glycoprotein required for viral entry and cell-to-cell spread, would restrict infection to a single round of viral replication and thus provide a safe, highly attenuated candidate vaccine. Therefore, we constructed a variant of HSV-2(G) deleted for $U_S6$ ($gD$-2) (*Cheshenko et al., 2013*). The virus was subsequently subjected to three rounds of plaque purification and grew to high titers ($10^8$ pfu/ml) when phenotypically complemented by growing the virus on VD60 cells, which express HSV-1 gD under the control of the $gD$-1 promoter. No plaques were observed when three independent plaque-purified isolates were grown to $10^8$ pfu on VD60 cells and then plated on non-complementing Vero cells. Moreover, no recombinants were detected by PCR, most likely reflecting differences in the regions flanking gD in VD60 cells and in HSV-2 (*Figure 1A*). Western blots show similar levels of expression of other HSV viral proteins when the deletion virus is grown on VD60 or Vero cells (*Figure 1B*).

To assess whether HSV-2 $\Delta gD^{-/+gD-1}$ is safe in vivo, severe combined immunodeficiency (SCID) mice were inoculated subcutaneously or intravaginally with the complemented HSV-2 $\Delta gD^{-/+gD-1}$ strain or parental HSV-2(G) virus (*Figure 2A*). SCID mice inoculated intravaginally with $10^7$ pfu of HSV-2 $\Delta gD^{-/+gD-1}$ manifested no signs of disease, whereas animals inoculated with $10^4$ pfu of HSV-2(G) quickly succumbed to the infection and manifested severe HSV-2-induced epithelial and neurological disease (*Figure 2B,C*). Subcutaneous inoculation with wild-type virus also induced disease (60% mortality with $10^5$ pfu), while no evidence of disease was observed following exposure to high doses of HSV-2 $\Delta gD^{-/+gD-1}$ in SCID mice. Moreover, no virus was detected in the neural tissue of SCID mice inoculated intravaginally with HSV-2 $\Delta gD^{-/+gD-1}$ by qPCR (*Figure 2D*).

To determine whether vaccination with HSV-2 $\Delta gD^{-/+gD-1}$ induces protective immunity to wild-type HSV-2 intravaginal challenge, mice were primed and boosted subcutaneously (sc) with $5 \times 10^6$ pfu (100 µl total volume) or with an uninfected VD60 cell lysate as a control. 3 weeks after the boost, the mice were challenged intravaginally with a clinical isolate of HSV-2 (strain 4674) (*Segarra et al., 2011*) equivalent to an $LD_{90}$ ($5 \times 10^4$ pfu/mouse) and 10 times the $LD_{90}$ ($5 \times 10^5$ pfu/mouse). Vaccinated mice were completely protected from lethal disease, whereas all of the control mice succumbed by day 8 post–challenge (*Figure 3A*). The majority of the vaccinated mice (14/20) showed no evidence of epithelial disease (*Figure 3B*) and none showed any signs of neurological disease (*Figure 3C*). The mice with epithelial signs recovered fully and HSV was detected by plaque assay in vaginal washes from vaccinated mice only on day 2 post–challenge but was cleared by day 4 post–challenge (*Figure 3D*). These findings suggest that the epithelial signs observed may reflect a local immune response and not ongoing viral replication. Moreover, no infectious virus was recovered from vaginal or neural tissue in HSV-2 $\Delta gD^{-/+gD-1}$-vaccinated mice at day 5 post-infection, but virus was readily detected from all of these sites in control-vaccinated mice (*Figure 3E*). Furthermore, we were unable to reactivate any virus from dorsal root ganglia (DRG) cultured ex vivo for 21 days (*Figure 3F,G*). In contrast, virus reactivated from DRG extracted from all control-vaccinated animals. In addition, HSV DNA ($U_S6$ gene) was not detected by qPCR in vaccinated mice (limit of detection 3 HSV-2 genome copies), but was detected in all controls (*Figure 3H*). Similar results were obtained using primers for a different viral gene, *UL30* (data not shown). These findings suggest that the vaccine prevents establishment of latency.

Mice were pretreated with medroxyprogesterone (MPA) to hormonally synchronize them prior to immunization in the efficacy studies. However, MPA can modulate innate and adaptive immune

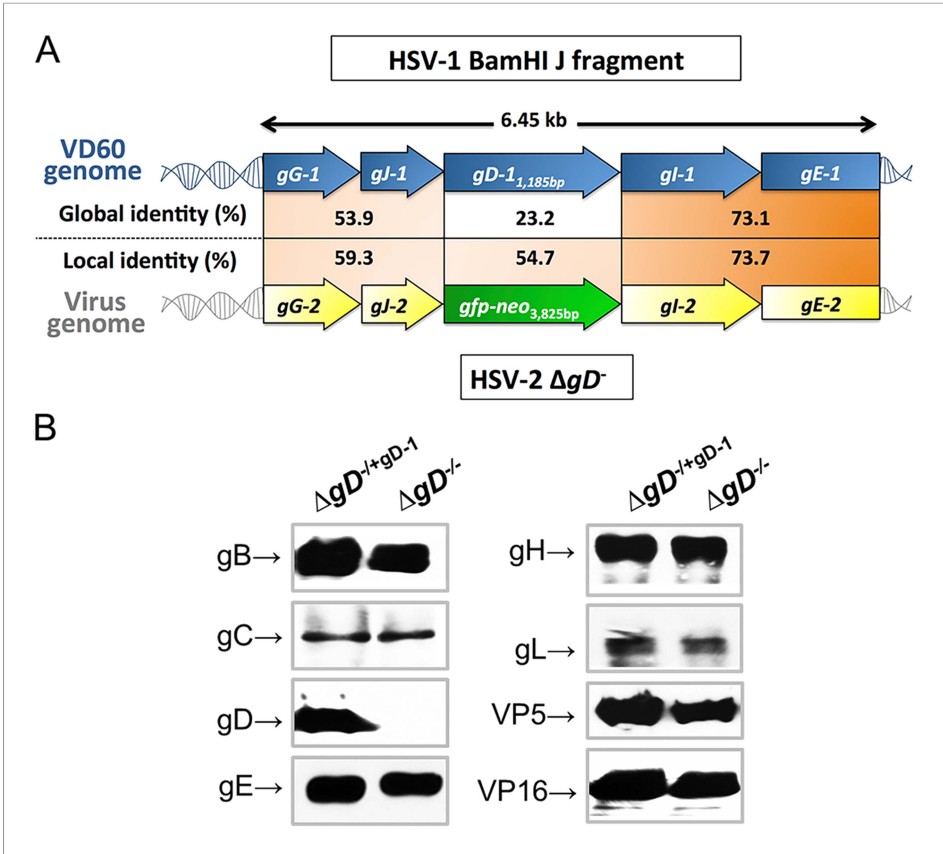

**Figure 1**. Characterization of the $\Delta gD^{-/-}$ virus. (**A**) Alignment of the upstream and downstream regions of *gD* located within the HSV-1 BamHI J fragment encoded in VD60 cells and within the genome of $\Delta gD^{-/-}$ using LALIGN (ExPASy) (*Myers and Miller, 1988*). Global alignments assess end-to-end sequences and local pairwise alignments search for regions with high identity. (**B**) Western blots of dextran gradient-purified virus isolated 24 hr after infection of VD60 ($\Delta gD^{-/+gD-1}$) and Vero ($\Delta gD^{-/-}$) cells. Protein expression was assessed for viral glycoproteins B (gB, $U_L27$), gC ($U_L44$), gD ($U_s6$), gE ($U_s8$), gH ($U_s22$), gL ($U_L1$), VP5 ($U_L19$) and VP16 ($U_L48$).

responses (*Kaushic et al., 2003*; *Vicetti Miguel et al., 2012*). Therefore we conducted additional experiments with cycling and MPA treated mice. The vaccine protected 100% of mice independent of hormonal treatment (*Figure 3—figure supplement 1*). In addition, mouse strains can differ in susceptibility and immune responses to pathogens. Therefore we also tested the efficacy of the vaccine against HSV-2 intravaginal infection in BALB/c mice. Consistent with the findings in C57BL/6 mice, 100% of HSV-2 $\Delta gD^{-/+gD-1}$-vaccinated mice survived intravaginal challenge (*Figure 3I*) and no infectious virus or viral DNA was recovered from neural tissues at day 7 and 21 post-infection (*Figure 3J*).

## HSV-2 $\Delta gD^{-/+gD-1}$ protects against disease in a skin scarification model

The majority of clinical HSV lesions are observed on the external genital skin rather than cervicovaginal sites. Therefore, we tested the vaccine for efficacy in a modified skin scarification model against both serotypes. The flank skin of C57BL/6 mice was depilated and scarified prior to direct inoculation of the skin with $5 \times 10^4$ pfu (HSV-2(4674)) or $1 \times 10^7$ pfu (HSV-1(17)) in 5 μl volume. The control-vaccinated (VD60 lysate) mice developed zosteriform lesions, neurologic disease and succumbed to infection following challenge with HSV-2, whereas HSV-1 produced more modest signs (*Figure 4A,B* and *Figure 4—figure supplement 1*). More importantly, the HSV-2 $\Delta gD^{-/+gD-1}$-vaccinated mice showed minimal signs of epithelial disease (maximal disease score 1), no neurological signs and 100% survival after challenge with either serotype (*Figure 4C* and *Figure 4—figure supplement 1*). No HSV was detected by titering or qPCR in day 14 skin biopsies or neural tissue from HSV-2

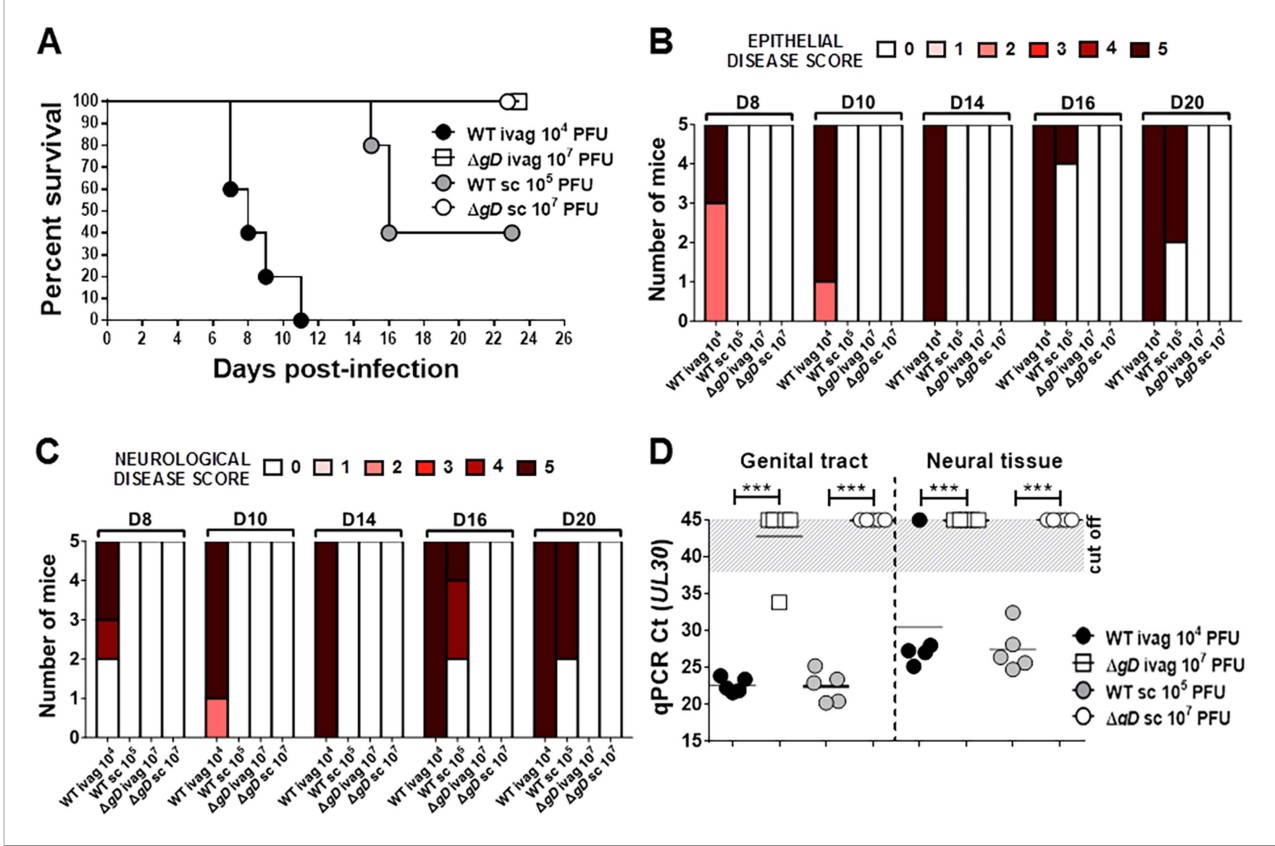

**Figure 2**. HSV-2 $\Delta gD^{-/+gD-1}$ is attenuated in severe combined immunodeficiency (SCID) mice. (**A**) Survival of SCID mice inoculated with up to $10^7$ pfu of HSV-2 $\Delta gD^{-/+gD-1}$ or up to $10^5$ pfu of the parental HSV-2(G) strain either intravaginally (ivag) or subcutaneously (sc). Statistical significance was measured by log-rank Mantel–Cox test; **p < 0.01 for $\Delta gD$ and WT after ivag inoculation. (**B**) Epithelial and (**C**) Neurological disease scores for SCID mice inoculated with the different viruses at indicated doses. (**D**) HSV-2 DNA (qPCR, $U_L30$ gene) in genital tract and neural tissue samples at day 5 post-virus inoculation. The Ct cut off was determined with HSV-uninfected naïve samples. Statistical significance was measured by two-way ANOVA with Sidak's multiple comparisons test for (**B**, **C** and **D**); ***p < 0.001. HSV-2 $\Delta gD^{-/+gD-1}$ and its parental strain are abbreviated as $\Delta gD$ and WT, respectively.

$\Delta gD^{-/+gD-1}$-vaccinated mice; whereas virus was detected in 100% of control mice challenged with HSV-2 (*Figure 4D,E*).

## Passive transfer of sera from immunized mice protects naïve mice from intravaginal and skin challenge

To determine if protection could be transferred to naïve mice, 250 µl of serum (containing 750 µg of immunoglobulin administered intraperitoneally) or $3 \times 10^6$ T cells (administered intravenously) isolated from the blood or spleen and lymph nodes of immunized mice, respectively, were administered to C57BL/6 mice. 48 hr later, the mice were challenged with $10^5$ pfu of HSV-2(4674) intravaginally. Mice treated with a single dose of immune serum displayed modest epithelial signs (mean score 2), but no neurologic disease with 100% survival (*Figure 5A* and *Figure 5—figure supplement 1A,B*). In contrast, mice that received T cells from immune mice or serum or T cells from the controls succumbed to disease. Depletion of immunoglobulins from the immune serum by passage on a Protein L column abolished the protective capacity. HSV-specific Abs were detected in vaginal washes in mice that received the immune, but not control serum (*Figure 5B*).

Intraperitoneal administration of serum also protected mice from skin disease. The immune sera treated mice survived challenge (*Figure 5C*) and developed moderate signs of epithelial disease (mean score of 2.2), which peaked on day 6 (*Figure 5—figure supplement 1C*). Moreover, no neurological signs were observed and no virus was detected in neural tissue by titering or qPCR (*Figure 5D,E*). In contrast, mice that received control serum developed severe epithelial and neurological disease and succumbed to the infection with 100% lethality.

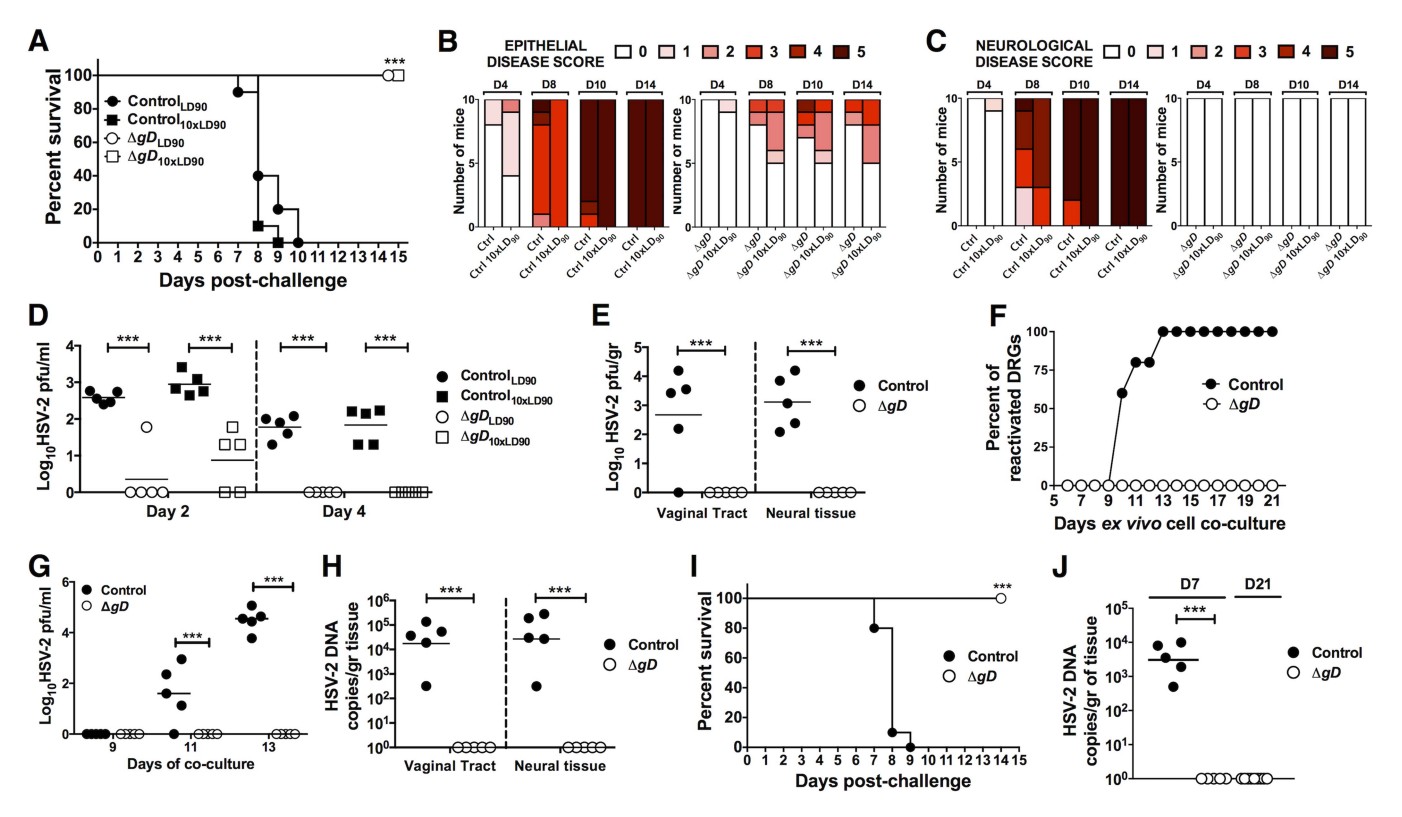

**Figure 3.** Vaccination with HSV-2 $\Delta gD^{-/+gD-1}$ protects mice against intravaginal lethal challenge. C57BL/6 mice were subcutaneously primed and boosted 3 weeks apart either with HSV-2 $\Delta gD^{-/+gD-1}$ or VD60 cell lysate (Control). 21 days after boost, mice were challenged with an $LD_{90}$ of wild-type HSV-2(4674) or $10 \times LD_{90}$. (**A**) Survival, (**B**) Epithelial and (**C**) Neurological disease scores were followed daily after challenge. (**D**) Viral titers in vaginal washes at days 2 and 4 after challenge (n = 10 mice pooled two per group, lines indicate means). (**E**) Viral titers in vaginal and neural tissue (including the dorsal root ganglia, DRG) at day 5 after challenge (n = 5, lines indicate means). (**F**) Ex vivo reactivation of neural tissue obtained from challenged mice (n = 5 per group). (**G**) HSV-2 pfu in the media of ex vivo reactivated neural tissue obtained from challenged mice (n = 5 mice per group, lines indicate means). (**H**) HSV-2 DNA (qPCR, $U_S6$ gene) of genital tissue and neural tissue at day 5 after challenge (n = 5, lines indicate means). (**I**) Survival of BALB/c mice that were primed, boosted and challenged as the C57BL/6 mice described above. (**J**) HSV-2 DNA (qPCR, $U_S6$ gene) in Control- and $\Delta gD$-vaccinated BALB/c neural tissue at day 7 (n = 5, lines indicate means) and $\Delta gD$-vaccinated BALB/c neural tissue at day 21 (n = 9, lines indicate means) after challenge. HSV-2 $\Delta gD^{-/+gD-1}$-vaccinated group vs Control-vaccinated group were compared by log-rank Mantel–Cox test (**A, I**), two-way ANOVA with Sidak's multiple comparisons test (**D, E, G, H**) or unpaired t-test (**J**); ***p < 0.001. HSV-2 $\Delta gD^{-/+gD-1}$ and Control are abbreviated as $\Delta gD$ and Ctrl, respectively.

The following figure supplement is available for figure 3:

**Figure supplement 1.** $\Delta gD^{-/+gD-1}$-vaccinated cycling mice are protected against intravaginal HSV-2 challenge.

## HSV-2 $\Delta gD^{-/+gD-1}$ induces high levels of antibodies targeting multiple HSV proteins with antibody-dependent cell-mediated cytotoxicity (ADCC) activity

HSV-specific Ab responses were consistently observed with serum dilutions as high as 1:800,000 7 days post-boost (**Figure 6A**) and were detected in vaginal washes within 4 days of challenge, indicating transport of HSV Abs into the vaginal lumen (**Figure 6B**). The serum HSV-specific Abs exhibited low levels of neutralizing activity (1:5 dilution) (**Figure 6C**), but did display ADCC activity, which was reduced in the presence of anti-FcγR Ab (**Figure 6D**). The HSV-specific Abs in serum and vaginal washes were comprised predominantly of IgG2a and IgG2b (**Figure 6E** and **Figure 6—figure supplement 1A**) and recognized multiple viral proteins on Western blots with HSV-2-infected cell lysates as the immunogen (**Figure 6F**). Serum from control-immunized and subsequently HSV-2 infected mice (obtained at day 8 post–challenge) recognized a more restricted set of proteins.

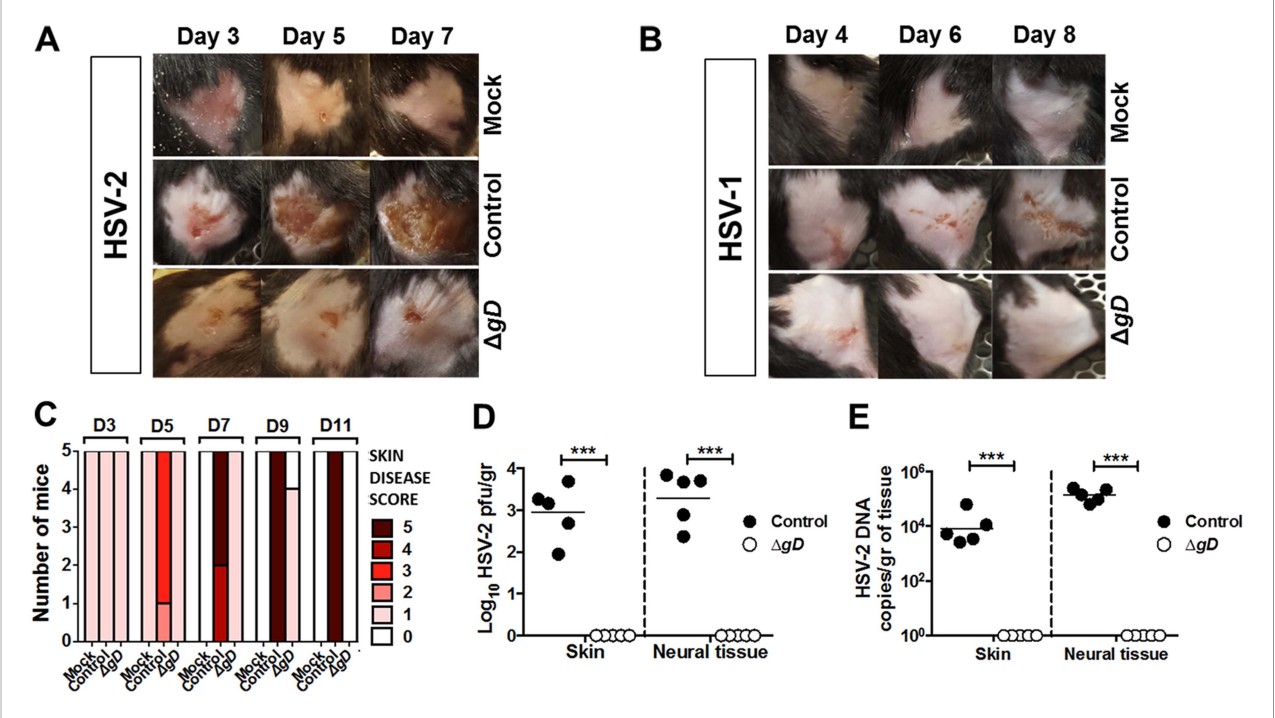

**Figure 4**. Vaccination with HSV-2 $\Delta gD^{-/+gD-1}$ protects mice infected with HSV-2 and HSV-1 in a skin scarification model. Mice were subcutaneously primed and boosted 3 weeks apart either with HSV-2 $\Delta gD^{-/+gD-1}$, Control VD60 cell lysate or PBS. 3 weeks later, mice were depilated and challenged in the flank skin with PBS (mock), (**A**) $5 \times 10^4$ pfu HSV-2(4674) or (**B**), $1 \times 10^7$ pfu HSV-1(17). Representative images are shown. (**C**) Skin disease scores for HSV-2(4674)-challenged mice at days 3–11. (**D**) Viral titers from biopsies of skin or neural tissue obtained on day 6–7 (Control mice) and day 14 (HSV-2 $\Delta gD^{-/+gD-1}$-vaccinated mice) (n = 5 mice per group, lines indicate means). (**E**) HSV-2 DNA (qPCR, $U_S6$ gene) in skin biopsies and neural tissue of Control mice (day 6–7) and HSV-2 $\Delta gD^{-/+gD-1}$-vaccinated mice (day 14) challenged with virus (5 mice per group, lines indicate means). Statistical significance was measured by two-way ANOVA with Sidak's multiple comparisons test (**D** and **E**); ***p < 0.001, $\Delta gD^{-/+gD-1}$-vaccinated group vs control group. HSV-2 $\Delta gD^{-/+gD-1}$ is abbreviated as $\Delta gD$.

The following figure supplement is available for figure 4:

**Figure supplement 1**. Vaccination with HSV-2 $\Delta gD^{-/+gD-1}$ protects mice infected with HSV-1 in a skin scarification model.

Western blots with purified HSV-2 $\Delta gD^{-/+gD-1}$ or HSV-2 $\Delta gD^{-/-}$ indicate that the predominant antigen recognized by control-immunized, HSV-2-challenged, but not $\Delta gD^{-/+gD-1}$-vaccinated mice, is gD (*Figure 6—figure supplement 1B*). Blots with recombinant glycoprotein B-1 identify the ~100 kDa band recognized by the $\Delta gD^{-/+gD-1}$-vaccinated immune serum as gB (*Figure 6—figure supplement 1C*).

Importantly, antibody-mediated protection was lost when serum was transferred to FcγR and FcRn knock-out mice (*Figure 7A,B* and *Figure 7—figure supplement 1A,B*). HSV-specific Abs were detected in vaginal washes of the FcγR$^{-/-}$ mice, but failed to provide protection, whereas Abs were not detected in vaginal washes from FcRn$^{-/-}$ mice consistent with the role of FcRn in transport of Abs from serum to the mucosa (*Li et al., 2011*) (*Figure 7C,D*).

## Discussion

The HSV vaccine field has focused most of its efforts on the induction of neutralizing Abs to gD as the major determinant of protection. The current study challenges this paradigm. We found that a live attenuated virus deleted in HSV-2 gD protects mice from vaginal, skin, and neuronal disease. Specifically, we observed no evidence of neurological disease in any model and could not detect virus in DRG of vaccinated mice by ex vivo plaque assays or by qPCR with a lower limit of detection of three viral genomes. The vaccine was effective in C57BL/6 and BALB/c mice and elicited high titers of HSV Abs (1:800,000) that were capable of passively protecting mice from both intravaginal and skin

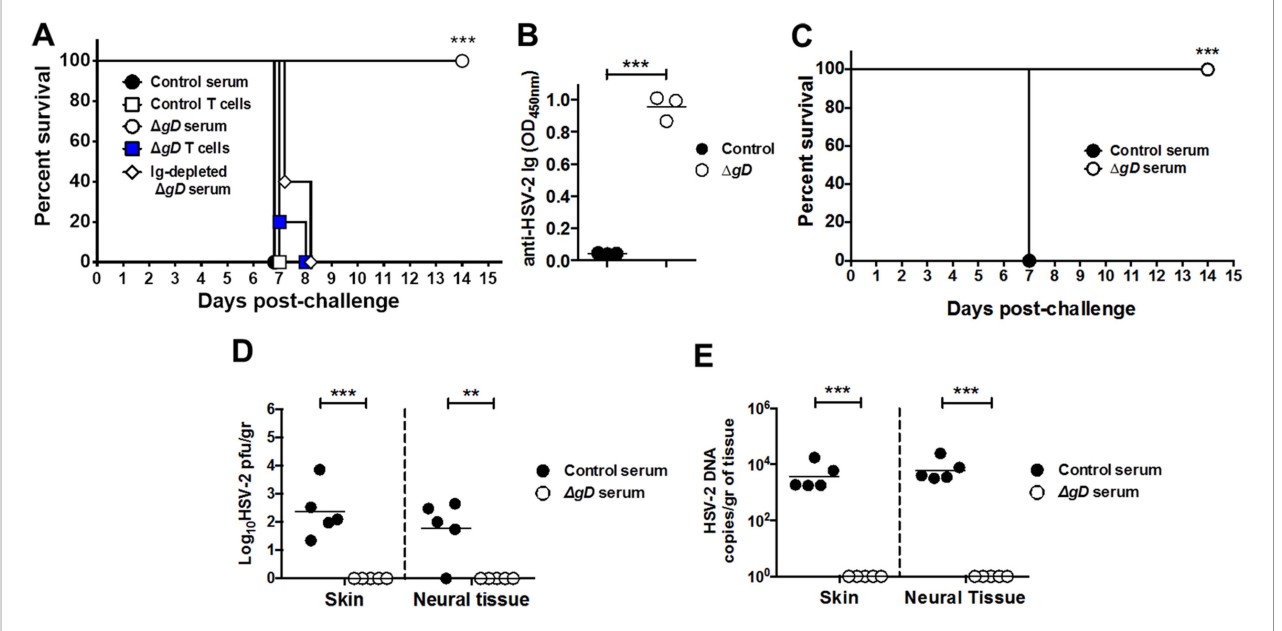

**Figure 5**. Serum from HSV-2 $\Delta gD^{-/+gD-1}$-vaccinated mice protects naïve mice against HSV-2 intravaginal and skin challenge. Mice were subcutaneously primed and boosted 3-weeks apart either with HSV-2 $\Delta gD^{-/+gD-1}$ or VD60 cell lysate (Control). 21 days later, blood and spleen were collected for serum and T cell purification and transferred intraperitoneally and intravenously, respectively, into naïve wild-type C57BL/6 mice. 24 hr and 48 hr after serum and T cell transfer, respectively, mice were challenged intravaginally with $LD_{90}$ of HSV-2(4674) and followed for survival (n = 5 mice per group). Serum immunoglobulins were depleted using a Protein L column (**A**). (**B**) Transferred anti-HSV-2-antibodies were assessed by ELISA in vaginal washes of recipient mice (washes pooled from five mice in three independent experiments). (**C**) Pooled serum from Control- or HSV-2 $\Delta gD^{-/+gD-1}$-vaccinated mice was transferred into naïve wild-type C57BL/6 mice. 24 hr after serum transfer, mice were depilated in the flank skin and challenged with HSV-2(4674) and followed for survival (n = 5 mice per group). (**D**) Viral titers in skin biopsies and neural tissue of mice receiving Control-serum (day 7) and $\Delta gD^{-/+gD-1}$-serum (day 14) (n = 5 mice per group). (**E**) HSV-2 DNA (qPCR, $U_S6$ gene) in skin biopsies and neural tissue of mice receiving Control-serum (day 7) and $\Delta gD^{-/+gD-1}$-serum (day 14) (n = 5 mice per group). Statistical significance was measured by log-rank Mantel–Cox test (**A** and **C**), t-test (**B**) and two-way ANOVA with Sidak's multiple comparisons test (**D**); **p < 0.01, ***p < 0.001, treatment vs control. HSV-2 $\Delta gD^{-/+gD-1}$ is abbreviated as $\Delta gD$.

The following figure supplement is available for figure 5:

**Figure supplement 1**. Serum from HSV-2 $\Delta gD^{-/+gD-1}$-vaccinated mice protects naïve mice against epithelial and neurological disease after HSV-2 intravaginal and skin challenge.

challenges. Notably, the Abs had low levels of neutralizing activity but did elicit ADCC and were able to passively transfer protection with a single dose of immune serum. The HSV-specific Abs could not be detected in vaginal washes of vaccinated mice, but were rapidly recruited into the vaginal lumen following intravaginal challenge and were associated with rapid clearance of the virus. Notably, the vaginal wash Abs were predominantly IgG, not IgA (data not shown), which is consistent with prior studies that have shown that while the predominant immunoglobulin by weight in the vaginal mucus of immune mice is secretory IgA, HSV-specific Abs are almost entirely IgG (*Parr and Parr, 1998*; *Parr et al., 1998*). The HSV-specific Abs were likely critical for protection as evidenced by the studies with FcRn knockout mice. Passive protection was also lost in FcγR knockout mice highlighting the role of Fc-mediated effector mechanisms.

These findings differ from those obtained with other subunit or attenuated vaccine candidates. The first attenuated vaccine to be evaluated clinically was a virus deleted in glycoprotein H (gH, $U_L22$). Unlike the data presented here with the HSV-2 $\Delta gD$ virus, the gH deletion viral vaccine did not prevent the establishment of latency in mice (*Speck et al., 1996*), was not shown to passively transfer protection and failed to reduce recurrences when evaluated as a therapeutic vaccine in a clinical trial (*de Bruyn et al., 2006*). More recently, a virus deleted in HSV-2 $U_L5$ and $U_L29$, which is defective in HSV DNA replication, has been advanced to a Phase 1 study. This deletion virus also differs from the HSV-2 $\Delta gD^{-/+gD-1}$ vaccine candidate described here in that it does not induce as high an Ab response,

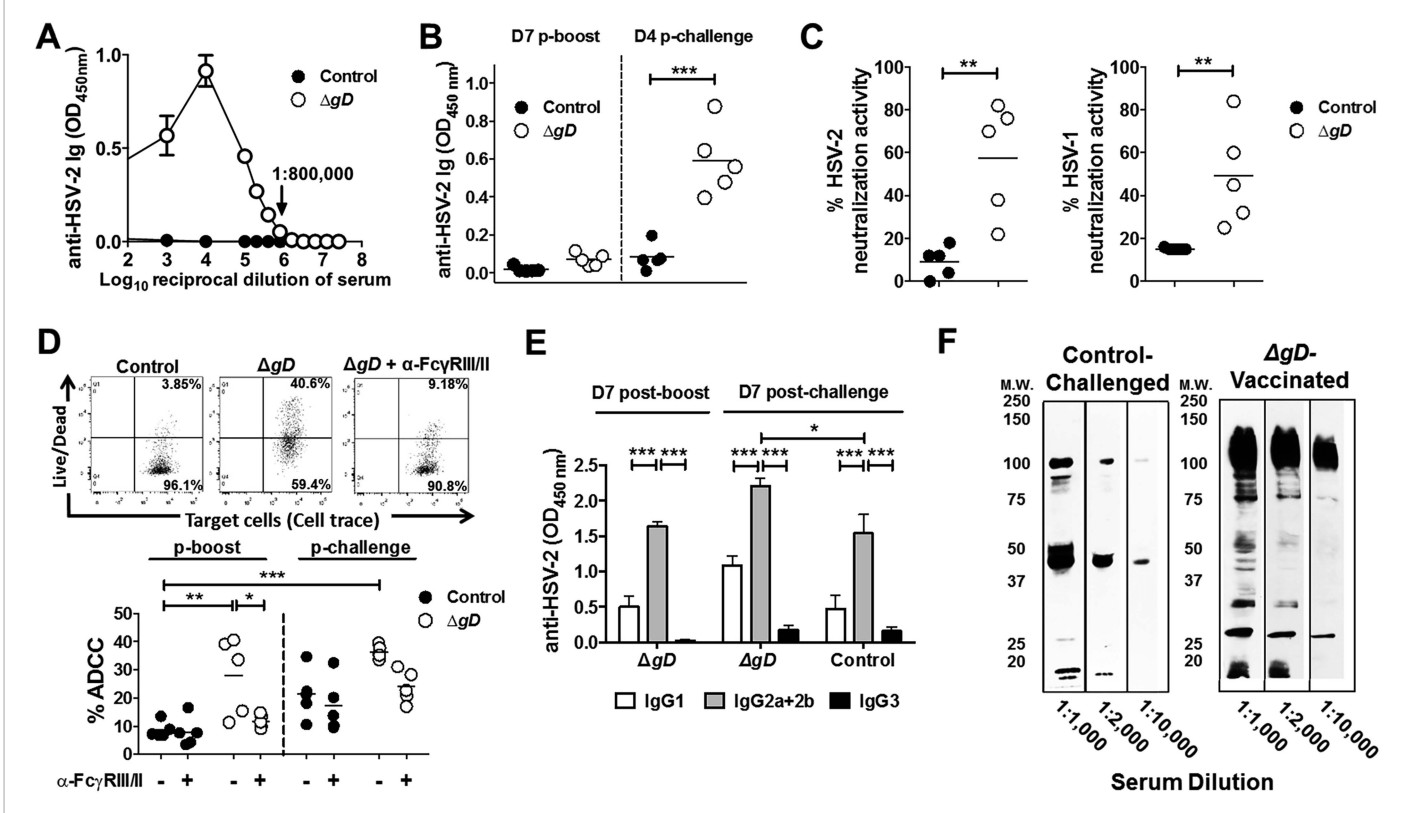

**Figure 6**. Vaccination with HSV-2 ΔgD⁻/+gD⁻¹ induces protective mucosal antibodies targeting multiple HSV proteins with ADCC activity. (**A**) Anti-HSV-2 antibodies detected by ELISA in serum samples at day 7 post-boost in mice subcutaneously primed and boosted 3-weeks apart with ΔgD⁻/+gD⁻¹ or VD60 lysate (Control) (4 independent pools of serum from 5–10 mice each, results shown as means ± SD). (**B**) Anti-HSV-2 antibodies detected by ELISA in vaginal washes at day 7 post-boost and day 4 post–challenge with HSV-2(4674) (n = 5 mice per group, lines indicate means). (**C**) In vitro neutralizing activity of serum antibodies (1:5 dilution) obtained from HSV-2 ΔgD⁻/+gD⁻¹- or Control-vaccinated mice against HSV-2 (left) and HSV-1 (right) (n = 5 mice per group, lines indicate means). (**D**) Antibody-dependent cell-mediated cytotoxicity (ADCC) using mouse splenocytes, HSV-2-infected Vero cells and serum obtained either from Control- (VD60 cell lysate) or HSV-2 ΔgD⁻/+gD⁻¹-vaccinated mice conducted in the absence or presence of anti-CD16/CD32 Ab to FcγRIII and FcγRII. % ADCC is defined as the percentage of dead (Live/Dead⁺) target cells within HSV-2 GFP^High positive cells. A representative dot blot is shown in the upper panel and lower panel shows results for five mice per group (lines indicate means). (**E**) Isotype of anti-HSV-2 serum antibodies obtained from five mice each that were either HSV-2 ΔgD⁻/+gD⁻¹-vaccinated and HSV-2(4674)-challenged or Control-vaccinated and HSV-2(4674)-challenged (results shown as means ± SD). (**F**) Western blots of cellular lysates infected with HSV-2(4674) and probed with dilutions of sera obtained from VD60 lysate-vaccinated and then subsequently infected mice (Control-Challenged) or dilutions of sera from HSV-2 ΔgD⁻/+gD⁻¹-vaccinated mice 7 days post boost (ΔgD-Vaccinated); blots are representative of five independent experiments. HSV-2 ΔgD⁻/+gD⁻¹-vaccinated groups were compared to control-vaccinated mice by two-way ANOVA with Sidak's multiple comparisons test (**A**, **B**, **D** and **E**) and unpaired t-test (**C**); *p < 0.05; **p < 0.01, ***p < 0.001. HSV-2 ΔgD⁻/+gD⁻¹ is abbreviated as ΔgD.

The following figure supplement is available for figure 6:

**Figure supplement 1**. Characterization of vaginal wash and serum antibodies.

has not been shown to passively protect, and did not prevent the establishment of latency as low levels of viral DNA were detected in the DRG following both immunization and viral challenge (*Hoshino et al., 2005*, *2008*, *2009*; *Dudek et al., 2008*). Notably, the $U_L5/U_L29$ deletion virus expresses gD at levels similar to that of wild-type virus, but lower levels of gB (*Da Costa et al., 2000*). Thymidine kinase (tk) deletion viruses, which have been used to study HSV immunogenicity because they are attenuated in mice (but not humans), express the full repertoire of viral proteins including gD. Protection following immunization with tk-deletion viruses requires anti-HSV-2 T cells in the genital tract (*Milligan et al., 2004*; *Shin and Iwasaki, 2012*) and serum from mice immunized with this

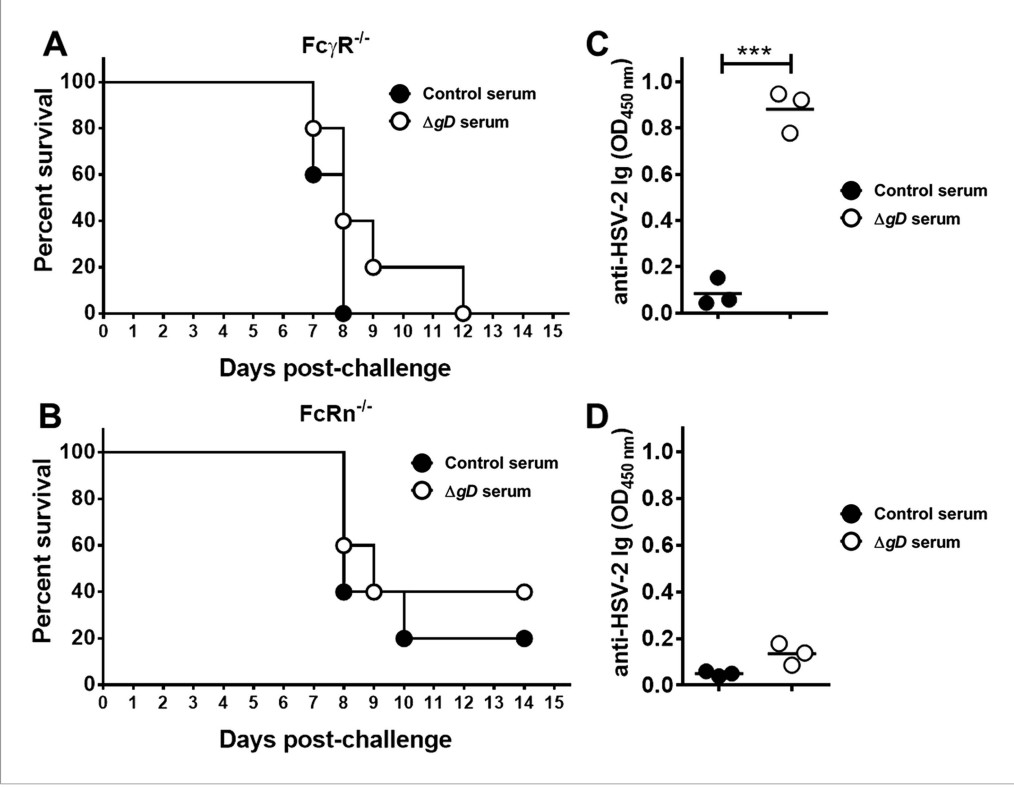

**Figure 7**. Antibody-mediated protection requires FcγR and FcRn expression. Survival of (**A**) FcγR$^{-/-}$ and (**B**) FcRn$^{-/-}$ mice that were either transferred serum obtained from Control- (VD60 cell lysate) or HSV-2 ΔgD$^{-/+gD-1}$-vaccinated wild-type mice and then challenged intravaginally with HSV-2(4674) (n = 5 mice per group). Detection of HSV-specific Abs by ELISA in pooled vaginal washes (n = 3 pools) of (**C**) FcγR$^{-/-}$ and (**D**) FcRn$^{-/-}$ mice receiving serum (intraperitoneally) from control- (VD60 cell lysate) or HSV-2 ΔgD$^{-/+gD-1}$-vaccinated wild-type mice. Survival curves were compared using log-rank Mantel–Cox test (**A** and **B**) and antibody titers using unpaired $t$-test (**C** and **D**); ***p < 0.001. HSV-2 ΔgD$^{-/+gD-1}$ is abbreviated as ΔgD.

The following figure supplement is available for figure 7:

**Figure supplement 1**. Antibody-mediated protection requires FcγR and FcRn expression.

tk-deletion virus has not been shown to passively transfer protection (*Morrison et al., 2001*; *Milligan et al., 2004*).

Possibly, deletion of gD, which is immunodominant as evidenced by the immune response in control, HSV-2 challenged mice (*Figure 6F*) and by clinical data showing that the majority of HSV-infected individuals have neutralizing Abs to gD (*Cairns et al., 2014*), unmasks viral antigens important in generating a protective humoral response. Additionally, gD may modulate immune responses through its interactions with herpesvirus entry mediator (HVEM) on immune cells (*La et al., 2002*; *Aubert et al., 2009*; *Stiles et al., 2010*; *Kopp et al., 2012*; *Grauwet et al., 2014*; *Sharma et al., 2014*). Interactions between gD and HVEM may skew the immune response towards neutralizing Abs whereas in the absence of the gD-HVEM interaction or yet undiscovered immunosuppressive functions of gD, the host may mount a polyantigenic IgG2 dominant response capable of eliciting FcR-mediated protection. This hypothesis will be tested in future studies by: (1) comparing, for example, the immune response to virus in mice deficient for HVEM, (2) examining the immune response to the HSV-2 ΔgD$^{-/+gD-1}$ vaccine when combined with the gD subunit vaccine, or (3) isolating mutant forms of gD that have lost immunosuppressive functions.

The inability of gD to elicit protective immunity against HSV-2 is highlighted by the results of the clinical trials with an adjuvanted gD subunit vaccine (*Mertz et al., 1990*; *Corey et al., 1999*; *Kohl et al., 2000*; *Bernstein et al., 2005*; *Belshe et al., 2012*). An additional mechanism that may

contribute to the protection observed with the HSV-2 $\Delta gD^{-/+gD-1}$ vaccine and to the observation that it induces Abs to multiple viral proteins is that by deleting the gene for gD, which is required for cell-to-cell spread, the cells initially infected by the complemented virus may function as a 'factory' generating viral proteins and 'defective' particles that are immunogenic. Consistent with this is the observation that heat-killed HSV-2 $\Delta gD^{-/+gD-1}$ failed to protect mice from vaginal challenge (data not shown).

In addition to its unique ability to generate high levels of Abs capable of translocating to mucosal sites and to prevent virus from reaching neuronal tissue following intravaginal or skin challenge, the HSV-2 $\Delta gD^{-/+gD-1}$ virus is also an attractive vaccine because of its safety. The virus induced no disease in SCID mice inoculated with 1000 times the lethal dose of wild-type virus and to date we have observed no recombinants either by plaque assay or PCR. This likely reflects that the virus is propagated on an HSV-1 and not an HSV-2 complementing cell line. The flanking regions of gD in HSV-1 and HSV-2 share approximately 54% and 73% identity within the 5′ and 3′ region of the $U_S6$ gene, respectively (HSV-1(KOS) vs HSV-2(HG52), NCBI genome, *Figure 1A*). This level of homology makes recombination highly unlikely.

It remains to be determined whether pre-existing immunity to HSV-1 will impact the immunogenicity of the HSV-2 $\Delta gD^{-/+gD-1}$ vaccine as was shown with the gD subunit vaccine (*Stanberry et al., 2002*) This may be less of a problem with HSV-2 $\Delta gD^{-/+gD-1}$ as it elicits a polyantigenic response whereas infection primarily elicits Abs to gD as evidenced by the immunoblots. If future studies suggest this is a problem, an alternative strategy is to complement the HSV-2 $\Delta gD^{-/+gD-1}$ virus with gD from a nonhuman herpes virus or a chimeric gD (*Zago et al., 2004*). Ideally the development of new mouse models of HSV latency and reactivation will allow us to explore the efficacy of HSV-2 $\Delta gD^{-/+gD-1}$ as a therapeutic vaccine.

The demonstration that this vaccine strain confers protective Abs against skin and intravaginal challenge that function through the Fc receptor highlights the importance of ADCC and antibody-mediated phagocytosis. Consistent with this concept are the results of the HIV RV144 vaccine trial, which found that non-neutralizing antibodies that mediate ADCC and antibody-mediated phagocytosis contributed to protection against HIV (*Bonsignori et al., 2012*). Similar findings have been observed in nonhuman primate studies of SHIV (*Florese et al., 2009*; *Bialuk et al., 2011*). Moreover, recent studies suggest that Ab–FcγR interactions are required to mediate protection against influenza (*Dilillo et al., 2014*). Together these findings suggest that the HSV-2 $\Delta gD^{-/+gD-1}$ vaccine provides a model to elicit such protective responses and indicate that it may be an ideal vaccine vector for other pathogens that enter through these sites.

## Materials and methods

### Mice

Experiments were carried out with approval from the Albert Einstein College of Medicine Institutional Animal Care and Use Committee, Protocol #20130913. Female C57BL/6 (wild-type), BALB/c (wild-type), severe combined immunodeficiency (SCID, BALB/c background), and FcRn$^{-/-}$ (B6.129X1-Fcgrt$^{tm1Dcr}$/DcrJ) mice were purchased from the National Cancer Institute (NCI, Frederick, MD) or Jackson Laboratory (JAX, Bar Harbor, ME) at 4–6 weeks of age. Female FcγR$^{-/-}$ (B6.129P2-Fcer1g$^{tm1Rav}$ N12) mice were purchased from Taconic (Albany, NY) at 4–6 weeks of age.

### Cell lines and viruses

CaSki (human cervical epithelial cell line; CRL-1550; American Type Culture Collection [ATTC], Manassas, VA, USA), Vero (green monkey kidney cell line; CCL-81; ATCC) cells, human keratinocytes (HaCAT, provided by David Johnson, Oregon Health & Sciences University, Portland, OR), and VD60 cells (Vero cells encoding gD-1 under the endogenous gene promoter [*Ligas and Johnson, 1988*]) were passaged in DMEM supplemented with 10% fetal bovine serum (FBS, Gemini Bio-Products, West Sacramento, CA). Splenocytes were obtained by generating a single cell suspension from harvested spleens from animals and subsequently, red blood cells were lysed with ACK lysis buffer (Gibco, Grand Island, NY). Splenocytes were then resuspended in complete RPMI media until use. Construction of the HSV-2 $\Delta gD^{-/+gD-1}$ virus in the HSV-2(G) (*Ejercito et al., 1968*) genetic background has been previously described by our group (*Cheshenko et al., 2013*). Briefly, 1,000 bp upstream and downstream regions of the $U_S6$ gene were PCR-amplified using viral genomic DNA (HSV-2(G)) with primers LL-V91I-US6 TTTTTT TTCCAT AAATTG GAAAGG GAACAG CGACCA AATGTC AC and LR-V91I-US6 TTTTTT TTCCAT TTCTTG GTGATA CGCGAT GCACAC GAAAAA CG for the upstream

region of $U_S6$ and primers RL-V91I-US6 TTTTTT TTCCAT AGATTG GTTCCC CGCTCC CGTGTA CC and RR-V91I-US6 TTTTTT TTCCAT CTTTTG GCGGGG GCGCCT GTATCG G for the downstream region of this gene. Stocks of HSV-2 $\Delta gD^{-/+gD-1}$ virus were propagated on complementing VD60 cells and titered on VD60 and Vero cells. HSV-2(4674) (*Nixon et al., 2013*) was propagated on HaCAT cells. HSV-1(17) (*Brown et al., 1973*), HSV-2(G) (*Ejercito et al., 1968*), HSV-1(F) (*Ejercito et al., 1968*) and HSV-2(333)ZAG (*Nixon et al., 2013*), a recombinant virus expressing the green fluorescent protein (GFP) were propagated on Vero cells. Concentrated viral stocks were stored at −80°C and diluted in PBS to the desired concentration when needed.

## Immunization, antibody passive transfer, and T cell adoptive transfer

5 days prior to immunization, mice were treated subcutaneously with 2.5 mg of medoxyprogesterone acetate (MPA; Sicor Pharmaceuticals, Irvine, CA). All prime immunizations were given subcutaneously (sc, medial to the hind limb and pelvis) with $5 \times 10^6$ pfu of HSV-2 $\Delta gD^{-/+gD-1}$ or an equal amount of VD60 cell lysate (Control). 3 weeks after initial immunizations, a boosting dose of $5 \times 10^6$ pfu of $\Delta gD^{-/+gD-1}$ or VD60 cell lysate was administered. For adoptive transfer of T cells, spleens were harvested from immunized mice 3 weeks after the boosting dose. Splenocytes obtained by single cell suspensions from harvested spleens were then treated with ACK Lysing buffer (Gibco) to remove erythrocytes and resuspended in MACs buffer (2% heat-inactivated FBS, 2 mM EDTA in PBS). T cells were magnetically purified via negative selection (Pan T Cell Isolation Kit II, Miltenyi Biotec, Germany) using LS columns. Before inoculation into recipient mice, purity and viability of cells were verified by flow cytometry and trypan blue staining. A total of $3 \times 10^6$ T cells were transferred into recipient mice intravenously through the tail. Alternatively, serum was collected from blood and a volume of 250 µl containing 750 µg of total IgG (detected via anti-mouse IgG ELISA) was transferred into recipient mice intraperitoneally 24 hr before intravaginal challenge of HSV-2. In select experiments, serum Ig was depleted by passing the serum through columns containing Protein L (GE healthcare Life Sciences, Piscataway Township, NJ).

## HSV-2 and HSV-1 infections

Mice were treated with MPA and challenged 5 days later with HSV-2(4674) 3 weeks after the vaccine boosting dose with $5 \times 10^4$ plaque forming units (pfu, LD$_{90}$) or $5 \times 10^5$ pfu (10× LD$_{90}$) of HSV-2(4674) intravaginally and monitored and scored for disease for 14 days as previously described (*Nixon et al., 2013*). For safety studies, unimmunized SCID mice were treated with MPA and inoculated 5 days later intravaginally or subcutaneously with HSV-2 (4674) or $\Delta gD^{-/+gD-1}$ either with 30 µl or 100 µl total volume, respectively. 150 µl of sterile PBS containing protease inhibitor (F Hoffmann-La Roche AG, Switzerland) was used to perform intravaginal washes for antibody and viral load quantification. Epithelial disease for intravaginal infections was scored as follows: (1) mild erythema, (2) hair loss, erythema, edema, (3) severe edema, hair loss, lesion formation, (4) severe ulcerations, multiple lesions and (5) death. Neurological disease for intravaginal infection was scored as follows: (1) urinary retention, (2) urinary retention and constipation, hind-limp paresis, (3) hind-limb paralysis (one leg), (4) complete hind limb paralysis (both legs), and (5) death. Mice were euthanized at a score of 3 or 4 and assigned a score of 5 on subsequent days for statistical analyses. For HSV skin infections, a protocol modified from *Goel et al. (2002)* was used. Briefly, mice were depilated on the right flank with Nair and allowed to rest for 24 hr. Depilated mice were anesthetized with isoflurane (Isothesia, Henry-Schein), then abraded on the exposed skin with a disposable emory board for 20–25 strokes and subsequently challenged with $5 \times 10^4$ pfu of HSV-2(4674) or $1 \times 10^7$ pfu of HSV-1(17) in 5 µl deposited on the abraded skin. Mice were then anesthetized for an additional 5 min to allow the virus inoculum to dry. Mice were monitored for 14 days and scored for signs of disease. Epithelial disease in the skin scarification model was scored as follows: (1) primary lesion or erythema, (2) distant site zosteriform lesions, mild edema/erythema, (3) severe ulceration and edema, increased epidermal spread, (4) hind-limb paralysis and (5) death. Mice that were euthanized at a score of 4 were given a value of 5 the next day.

## Virus detection in vaginal washes and tissue and ex-vivo reactivation in neural tissue

For pfu detection in tissue samples of mice inoculated and/or challenged with HSV, genital tract, skin, or dorsal root ganglia (including sciatic nerve from hind limb to spinal cord) were weighed and homogenized in serum-free Dubecco's Modified Eagles Medium (DMEM, Gibco) using RNase-free

pestles. Homogenized tissues were then sonicated for 30 s at maximum strength and centrifuged at 10,000×g for 5 min. Supernatants were then overlaid on confluent Vero cell monolayers ($2 \times 10^5$ cells/ well in a 48-well plate) for 1 hr. Wells were washed with PBS, then 199 medium (Gibco) containing 1% heat-inactivated FBS and overlaid with 0.5% methylcellulose and incubated at 37°C for 48 hr. Cells were fixed with 2% paraformaldehyde, stained with a crystal violet solution and the number of plaque forming units (pfu) were quantified. Vaginal washes from mice obtained after virus inoculation were plated neat or diluted in PBS on confluent Vero cell monolayers ($2 \times 10^5$ cells/well in a 48-well plate) and viral pfu were quantified as described above. Data are presented as $\log_{10}$ pfu per gram of tissue. For ex-vivo HSV reactivation, neural tissue obtained at day 5 post–challenge was excised, cut into 3–4 pieces and cultured with confluent Vero cell monolayers in serum-free DMEM in $60 \times 15$ mm tissue culture dishes (Corning). Dishes were observed daily up to 21 days post co-culture for cytopathic effect with media exchanged every 2 days. Supernatants were harvested every other day to measure viral pfu by standard plaque assay.

## HSV-2 RT-qPCR analysis

DNA was extracted from weighed tissue samples using DNeasy Blood and Tissue (Qiagen) following the manufacturer's recommendations. Extracted DNA was then normalized to 10 ng of DNA per reaction and viral DNA quantified using real-time quantitative PCR (RT-qPCR, qPCR) using ABsolute qPCR ROX Mix (Thermo Scientific). Primers for HSV-2 gD ($U_S6$) and polymerase ($U_L30$) were purchased from Integrated DNA Technologies ($U_S6$ cat: 117920498 and $U_L30$ cat: 1179200494) and used to detect viral genome DNA. To determine the number of HSV-2 genome copies, HSV-2 viral DNA was used as a standard curve (*Cheshenko and Herold, 2002*). Samples that read three or less copy numbers were considered negative. Data are presented as $\log_{10}$ HSV-2 copy numbers per gram of tissue.

## Antibody detection by ELISA

Serum was obtained from individual mice at 1 week post-prime, at boost, 3 weeks post-boost, and 8 days after intravaginal challenge with HSV-2. Vaginal washes were collected in 150 µl PBS containing protease inhibitor at different time points after sc boost with $\Delta gD^{-/+gD-1}$ or intravaginal challenge with HSV-2. For HSV-2 specific antibody detection, Vero cells were mock-infected or infected with HSV-2 (4674) and 24 hr later sonicated for 30 s and then coated on 96 well MaxiSorp ELISA plates (Nunc, NY) at a concentration of 45 µg cell lysate/well at 4°C overnight. Cell lysates were then further permeabilized with PBS/0.1% Triton X-100 buffer and fixed with 1% formaldehyde. Serum and vaginal washes obtained from individual mice were serially diluted and overlaid in duplicates onto cell lysate-coated wells and incubated at RT for 2 hr. Wells were washed with PBS/0.05% Tween 20 buffer five times and incubated with purified biotin anti-mouse Ig κ or biotin anti-mouse IgG1, IgG2a, IgG2b, or IgG3 at 1 µg/ml (Becton Dickenson, San Diego) for an additional 2 hr at RT. Next, cell lysates were washed as above and incubated with HRP-conjugated streptavidin (Becton Dickenson, San Diego) for 30 min at RT and then developed with TMB substrate (BD OptEIA) for 5 min. The reaction was stopped with 2 N $H_2SO_4$. Wells were read on a SpectraMax (M5 series) ELISA plate reader at an absorbance of 450 nm. The resulting absorbance was determined by subtracting values obtained for uninfected cell lysates to values obtained with infected cell lysates at a serum dilution of 1:2500 or 1: 100 (total Ig or isotype specific, respectively) and at a 1:5 dilution for vaginal washes.

## Neutralization assay

Serial dilutions of control and immune serum from 1 week post-boost were inactivated (56°C for 30 min) and then incubated at a 1:1 ratio with 50 pfu of HSV-2(4674) or HSV-1(F) in a 96-well plate at 37°C for 1 hr. Virus-antibody complexes were then overlaid onto confluent Vero cells in a 48-well plate for 1 hr at 37°C. Wells were then washed with PBS and subsequently with 500 µl 199 medium (Gibco) containing 1% heat-inactivated FBS with 0.5% methyl cellulose and incubated for 37°C for 48 hr. Plaques were quantified and percent neutralization was determined as the reduction of HSV-2 plaques compared to control (50 pfu HSV-2 in media).

## Antibody dependent cellular cytotoxicity assay

Vero cells (used as target cells) were infected with HSV-2:GFP(333)ZAG at an MOI of 1 pfu/cell. 18 hr post-infection cells were harvested via CellStripper (Corning) and labeled with CellTrace Violet

(Invitrogen, Grand Island, NY) at 1.25 μM. Labeled infected cells were then incubated with heat-inactivated serum from control or vaccinated individual mice at a 1:2 dilution for 15 min. Splenocytes from naïve mice were treated with ACK Lysing Buffer (Gibco) to remove erythrocytes and used as effector cells. Where indicated, effector cells were either treated or not with FCR-4G8 (anti-FcγRIII/II, anti-mouse CD16/32, Invitrogen, Grand Island, NY) at 2 μg/$10^6$ cells. Co-cultures were performed at an effector cell to target cell ratio of 25:1 for 4 hr. Effector/target and target cells were then stained with Live/Dead Red fixable dye (Invitrogen) for 15 min and subsequently washed and resuspended in FACs buffer (2% heat-inactivated FBS, 2 mM EDTA in PBS). The percentage of dead HSV-2-infected target cells was determined on a 5-laser LSRII flow cytometer (Becton Dickenson, San Diego, CA) at the Einstein Flow Cytometry Core Facility. Final analysis was carried out using FlowJo software version 10 (Tree Star, Inc., Ashland, OR).

## Western blots

Stocks of HSV-2 $\Delta gD^{-/+gD-1}$ and $\Delta gD^{-/-}$ were propagated on VD60 or Vero cells and purified using dextran or sucrose gradients as previously described (Cheshenko et al., 2013) and then analyzed for viral protein expression by performing Western blots. Equivalent numbers of viral particles (based on Western blot probing with antibody to the viral capsid protein, VP5 [Cheshenko et al., 2007]) were loaded on gels, proteins separated by SDS-PAGE, transferred to nitrocellulose and immunoblotted with anti-HSV-2; gC (H1196), anti-gE (101), anti-VP16 (VA16), anti-VP5 (3B6), anti-gB (10B7), anti-gD (0191) from Santa Cruz Biotechnology (Santa Cruz, CA) and anti-gH-gL (CH31, gift from R Eisenberg and G Cohen, University of Pennsylvania) as previously reported (Cheshenko et al., 2014). Serum immune-target profiling was done by performing Western blots with 10 μg of HSV-2(4674) infected Vero cell lysates per lane or equivalent particle numbers (based on Western blots for VP5) of HSV-2 $\Delta gD^{-/+gD-1}$ and $\Delta gD^{-/-}$ purified virus, and probing with serial dilutions of immune or control serum obtained 1 week post-boost or 8 days post-intravaginal challenge, respectively. Alternatively, Western blots were performed using 2 μg of recombinant glycoprotein B-1 produced in baculovirus and purified on a CL-68 sepharose column (17-0467-01, GE Healthcare, Sweden) as previously reported (Cheshenko et al., 2004). The anti-gB-1/gB-2 monoclonal antibody 10B7 and the anti-gD-1 monoclonal antibody 0191 (Santa Cruz Biotechnology) were used as positive controls. Blots were incubated with immune and control serum for 16 hr at 4°C. After repetitive washes, bound antibodies were detected using goat anti-mouse-HRP secondary antibody (Bio Rad, Hercules, CA).

## Statistical analysis

Results were compared by two-way analysis of variance (two-way ANOVA) with multiple comparisons or unpaired $t$ tests using GraphPad Prism version 6 (San Diego, CA). Mantel–Cox survival curves were compared by log rank tests. p values <0.05 (*), <0.01 (**), <0.001 (***) were considered significant.

## Acknowledgements

We are grateful to Mr. John Kim for assistance with experimental animals. We would also like to thank Kayla Weiss, Joern Schmitz, and Birgit Korioth-Schmitz for careful reading of this manuscript. The authors declare no financial conflict of interest.

## Additional information

### Funding

| Funder | Grant reference number | Author |
| --- | --- | --- |
| National Institutes of Health (NIH) | AI065309 | Christopher Petro, Thomas Jandl, Nazanin Khajoueinejad, Betsy C Herold |
| National Institutes of Health (NIH) | AI03461 | Natalia Cheshenko, Betsy C Herold |
| National Institutes of Health (NIH) | AI084225 | Thomas Jandl, Nazanin Khajoueinejad, Betsy C Herold |

| Funder | Grant reference number | Author |
|---|---|---|
| National Institutes of Health (NIH) | AI063537 | Christopher Petro, Angèle Bénard |
| Howard Hughes Medical Institute (HHMI) | Investigator | William Jacobs R Jr |
| Howard Hughes Medical Institute (HHMI) | Research Fellow | Pablo A González, Mayami Sengupta |

The funders had no role in study design, data collection and interpretation, or the decision to submit the work for publication.

## Author contributions

CP, PAG, Conception and design, Acquisition of data, Analysis and interpretation of data, Drafting or revising the article; NC, TJ, NK, MS, Acquisition of data, Analysis and interpretation of data; AB, Conception and design, Acquisition of data, Analysis and interpretation of data; BCH, WRJ, Conception and design, Drafting or revising the article

## Ethics

Animal experimentation: This study was performed in strict accordance with the recommendations in the Guide for the Care and Use of Laboratory Animals of the National Institutes of Health. All of the animals were handled according to approval from the Albert Einstein College of Medicine Institutional Animal Care and Use Committee, Protocol #20130913.

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
