## [Decision Letter]

Thank you for sending your work entitled “Herpes simplex type 2 virus deleted in glycoprotein D protects against vaginal, skin and neural disease” for consideration at *eLife*. Your article has been favorably evaluated by Randy Schekman (Senior editor) and 3 reviewers, one of whom served as a guest Reviewing Editor.

Two of the three reviewers, Susan Pierce and Andrew McMichael, have agreed to reveal their identity.

The Reviewing editor and the other reviewers discussed their comments before we reached this decision, and the Reviewing editor has assembled the following comments to help you prepare a revised submission.

The three reviewers were very positive regarding this manuscript. The six (major) points raised by reviewer #1 should be addressed by the authors. This need not involve additional experimentation. Finally, please discuss the points brought up by reviewers #2 and #3. No new experiments are required.

Reviewer #1

In their manuscript, Petro and colleagues describe a novel HSV-2 vaccine concept based on a gD deletion mutant. The authors show that the vaccine strain is only able to undergo a single cycle of replication in non-complementing cells and is safe in SCID mice at high doses. Furthermore, the construct protects from HSV-2 induced neurological and epithelial disease in mice and cross-protection in a skin scarification mouse model against HSV-1 is demonstrated as well. Interestingly, the authors identify ADCC as mechanism of protection. The study is important, innovative and certainly well done. However, there are some issues that require the authors’ attention.

Major points:

1) A live attenuated VZV vaccine (Varivax/Zostervax) is currently used in the US. Although this vaccine is highly efficacious the immune response is short-lived and re-immunization is required. The authors show protection after a relatively short time interval (3 weeks post boost). Do the authors foresee the same problem with their HSV-2 vaccine? How long lasting is the immune response?

2) The vaccine construct carries a HSV-1 gD protein. Approximately 54% of the US population has pre-existing immunity against HSV-1 (Bradley et al., JID, 2013). This could strongly impact on vaccine efficacy in humans. Would a complementing gD from a non-human herpesvirus be a better choice? The authors should discuss this.

3) The authors show crossprotection/crossneutralization against HSV-1. Is this crossprotection (Figure 4 and Figure 6) induced by the HSV-1 gD on the virus surface or not? No quantitative data is provided in Figure 4 for the HSV-1 challenge. The authors should add disease scores and virus titers.

4) The authors measure IgG in vaginal washes and use a neonatal FcR KO model to show that mucosal antibody is important. However, mucosal IgA is not discussed. Was that measured? Which role does it play?

5) In Figure 5, the authors show that passive transfer of sera from vaccinated mice protects naive mice from challenge. The authors include a group of mice that receive 'Ig' depleted serum. Animals in this group survive three days longer than mice in the other control groups and this result is likely statistically significant. Is this a technical issue with Ig depletion?

6) The authors show that the deletion of the immunodominant gD from their virus re-directs the immune response towards the less immunogenic gB, a concept that has also been proposed for HA stalk-based influenza virus vaccines. Despite high titers of antibodies against gB, the neutralization titers are very low, which is in contrast to findings with gB subunit vaccines which usually induce robust neutralization titers. The authors should discuss this.

Reviewer #2:

In this manuscript, the authors explore the efficacy of using a novel mechanism to attenuate Herpes simplex virus (HSV) 2 for use as a vaccine. HSV infections are an enormous global health problem and at present there is no HSV vaccine. Consequently new approaches to vaccine design are badly needed. The authors attenuate HSV-2 by deleting glycoprotein D (gD-2) and complementing the loss by a single round of replication of the virus on cells expressing HSV-1 gD (ΔgD^-/+gD-1^). The ΔgD^-/+gD-1^ virus when used as a vaccine should only replicate once in the host. To my mind the results presented here are remarkable. Subcutaneous immunization with ΔgD^-/+gD-1^ completely protected against intravaginal or skin challenge and prevented the establishment of latency. These data are convincing. Concerning the immune mechanisms underlying protection the authors provide evidence that protection can be transferred by Ig from immunized mice and requires that the recipient mouse express FcgammaR (presumably to mediate antibody-dependent cellular cytotoxicity as the Ig is poorly neutralizing but has ADCC activity) and FcRn (presumably to mediate transfer of Ig into the vaginal lumen).

The puzzling result in terms of immune mechanism is the dramatic effect of the presence of gD-1 on the Ab response. When gD is expressed the Ab response is almost entirely gD-specific and the Abs are neutralizing. In the absence of gD antibodies specific for many HSV proteins are made. This may simply be due to an extreme immunodominance of gD. It would seem that achieving such dominance would require that gD-specific B cells have a strong advantage over B cells recognizing other HSV antigens in germinal center reactions. Perhaps immune cells, for example, T follicular helper cells that have receptors for gD, synergize with gD-specific B cells. One way to approach the question would be to determine if gD expressing HSV induces only gD-specific response in mice deficient for the known gD receptors, i.e. CD155, CD112. Such experiments, while of interest, are certainly not required to support the conclusions drawn here.

Reviewer #3:

This manuscript describes a novel approach to HSV2 vaccination in mice. They find that deletion of the immunodominat gD protein in the virus elicits non-neutralizing antibodies which protect against challenge very impressively. They use a variety of readouts, all giving the same answer.

They show that the protection is mediated by antibody, because passive transfer protects though not if Ig depleted. They show that the protection is Fc dependent and go on to show that it is likely to act through ADCC. Whether this is the direct mechanism responsible is less clear but there is at least a good correlation between this assay in vitro and protection.

There are important general messages here. Neutralizing antibody is not essential for protection. That is known, though often ignored, in other virus infections; here it is very well illustrated and shown that such antibodies can be harnessed by vaccination to give protection. Secondly the immunodominant antigen may not elicit protective antibody but subdominant immunogens might; genetically manipulated vaccines offer a way to address that, though what determines immunodominance often remains a mystery.

Can the findings be generalized to other infections? They make a reference to the RV144 HV vaccine trial where some protection was given by non-neutralizing antibodies. It would be good to cite other more convincing examples, if they exist.

A small point that is relevant to human vaccinology. BALB/c and C57B6 mice are often used to test immunogens and for challenges. However in human terms these are just two individuals and may well be immunologically atypical compared to outbred mice or outbred humans. It would be good to test this protection in an outbred mouse model, if the same level of protection is seen that be very encouraging for a human vaccine. However this is not essential for acceptance of this manuscript.

---

## [Author Response]

Reviewer #1:

1) A live attenuated VZV vaccine (Varivax/Zostervax) is currently used in the US. Although this vaccine is highly efficacious the immune response is short-lived and re-immunization is required. The authors show protection after a relatively short time interval (3 weeks post boost). Do the authors foresee the same problem with their HSV-2 vaccine? How long lasting is the immune response?

Studies are in progress to test the durability of the immune response in mice. We recognize that as we advance into clinical trials, additional booster doses may be needed. It should be noted however, that the HSV vaccine candidate proposed in these studies was engineered with a specific defined deletion of the gene encoding the glycoprotein D. In contrast, the Varivx/Zostervax vaccine was attenuated through serial passages in human and guinea pig cells and was sequenced after introduction into the clinic. The specific determinants of attenuation are still uncertain. The open reading frame with the most single nucleotide polymorphisms encodes the regulatory protein IE62, but IE62 studies have failed to define a specific polymorphism associated with attenuation. A more recent next-generation sequencing study of an unrelated highly attenuated VZV strain identified a stop codon mutation in ORF0 (herpes simplex virus UL56 homolog) identical to one found in vOka, suggesting that this may be one of the determinants of the attenuation genotype of live VZV vaccines (J Virol, 2012, 86(19): 10695-703). This will be addressed in future studies.

*2) The vaccine construct carries a HSV-1 gD protein. Approximately 54% of the US population has pre-existing immunity against HSV-1 (Bradley et al., JID, 2013). This could strongly impact on vaccine efficacy in humans. Would a complementing gD from a non-human herpesvirus be a better choice? The authors should discuss this*.

Future studies will address whether HSV-1 seropositivity limits the immunogenicity of the vaccine strain or whether the vaccine would be optimal in younger children who are seronegative for both HSV-1 and HSV-2. To experimentally explore this, mice infected via the skin scarification model with HSV-1 and confirmed to be seropositive will be vaccinated and challenged intravaginally with HSV-2 or vice versa. This concern will also be addressed early in the clinical development by testing the safety and immunogenicity of the vaccine in HSV-2 seronegative subjects who are either HSV-1 seropositive or seronegative. If preexisting immunity to HSV-1 interferes with the vaccine immunogenicity, we will explore complementing the vaccine strain with a gD homologue from a non-human herpesvirus as suggested by the reviewer. Other non-human herpesviruses share partial homology with HSV gD. For example, pseudorabies virus (PrV), which also binds to nectin-1 as an entry receptor, shares 30% amino acid identity and did not substitute for HSV gD in a fusion assay. However, chimeric gD molecules composed of HSV and PRV sequences did substitute, provided the first 285 amino acids were from HSV-1 (PNAS, 2004, 101(50): 17498–17503). If needed, we will explore these alternative strategies. We have added a comment in the Discussion section indicating the need to explore the impact of preexisting HSV-1 seropositivity on the immunogenicity of this vaccine strain.

*3) The authors show crossprotection/crossneutralization against HSV-1. Is this crossprotection (*Figure 4
*and*
Figure 6*) induced by the HSV-1 gD on the virus surface or not? No quantitative data is provided in*
Figure 4
*for the HSV-1 challenge. The authors should add disease scores and virus titers*.

It is unlikely that the cross-protection against HSV-1 in the intravaginal and skin scarification models is mediated by antibodies to gD-1 as the immunoblots indicate only low levels of anti-gD antibodies (Figure 6). We speculate that antibodies targeting other viral antigens such as glycoprotein B contribute to the cross protection. This notion is supported by the observation that a humanized antibody to gB provided cross protection against HSV-1 and HSV-2 (PNAS, 2013, 110(17), 6760–5). Identification of the antigenic targets and subsequent depletion of antigen specific antibodies from the immune serum will enable us to identify the correlates of protection against both HSV-1 (cross-protection) and HSV-2. These studies will be conducted in the future. In addition, we plan to extend studies and test vaccine efficacy against clinical isolates of both HSV-1 and HSV-2. As suggested by the Reviewer, we have added the HSV-1 disease scores and skin viral loads to the manuscript (Figure 4—figure supplement 1).

4) The authors measure IgG in vaginal washes and use a neonatal FcR KO model to show that mucosal antibody is important. However, mucosal IgA is not discussed. Was that measured? Which role does it play?

We previously measured HSV specific IgA in vaginal washes but found low abundance in our ELISA assay. Although the predominant immunoglobulin by weight in the vaginal mucus of immune mice has been shown to be secretory IgA, prior studies (consistent with our work) indicate that virus specific antibody is almost entirely IgG (Parr, J Reproductive Immunology, 1998, 38; 15-3; and Parr, M., 1998, Journal of Virology, 72(6), 5137–45). Thus, we focused on IgG. It should also be noted that in humans, the primary vaginal antibody is IgG, not IgA. We have added a sentence and these references to the revised Discussion.

*5) In*
Figure 5
*the authors show that passive transfer of sera from vaccinated mice protects naive mice from challenge. The authors include a group of mice that receive 'Ig' depleted serum. Animals in this group survive three days longer than mice in the other control groups and this result is likely statistically significant. Is this a technical issue with Ig depletion?*

We thank the reviewer for noting this as it reflected an error in the uploading of the figure between different programs. There is no significant difference between the ‘Ig’ depleted group and the control groups and the figure has been corrected.

*6) The authors show that the deletion of the immunodominant gD from their virus re-directs the immune response towards the less immunogenic gB, a concept that has also been proposed for HA stalk-based influenza virus vaccines. Despite high titers of antibodies against gB, the neutralization titers are very low, which is in contrast to findings with gB subunit vaccines which usually induce robust neutralization titers. The authors should discuss this*.

We hypothesize that deletion of the immunodominant gD from the virus not only redirects the immune response towards other antigens including gB but also modulates the type of antibody response from Abs that elicit neutralization to those capable of eliciting ADCC and other FcR mediated functions. This may reflect the isotype and other characteristics and may be linked to the absence of the immunomodulating effects of the gD-HVEM interaction. We have expanded the Discussion to address this.

Reviewer #2:

*In this manuscript, the authors explore the efficacy of using a novel mechanism to attenuate Herpes simplex virus (HSV) 2 for use as a vaccine. HSV infections are an enormous global health problem and at present there is no HSV vaccine. Consequently new approaches to vaccine design are badly needed. The authors attenuate HSV-2 by deleting glycoprotein D (gD-2) and complementing the loss by a single round of replication of the virus on cells expressing HSV-1 gD (*ΔgD^-/+gD-1^*). The* ΔgD^-/+gD-1^
*virus when used as a vaccine should only replicate once in the host. To my mind the results presented here are remarkable. Subcutaneous immunization with* ΔgD^-/+gD-1^
*completely protected against intravaginal or skin challenge and prevented the establishment of latency. These data are convincing. Concerning the immune mechanisms underlying protection the authors provide evidence that protection can be transferred by Ig from immunized mice and requires that the recipient mouse express FcgammaR (presumably to mediate antibody-dependent cellular cytotoxicity as the Ig is poorly neutralizing but has ADCC activity) and FcRn (presumably to mediate transfer of Ig into the vaginal lumen)*.

*The puzzling result in terms of immune mechanism is the dramatic effect of the presence of gD-1 on the Ab response. When gD is expressed the Ab response is almost entirely gD-specific and the Abs are neutralizing. In the absence of gD antibodies specific for many HSV proteins are made. This may simply be due to an extreme immunodominance of gD. It would seem that achieving such dominance would require that gD-specific B cells have a strong advantage over B cells recognizing other HSV antigens in germinal center reactions. Perhaps immune cells, for example, T follicular helper cells that have receptors for gD, synergize with gD-specific B cells. One way to approach the question would be to determine if gD expressing HSV induces only gD-specific response in mice deficient for the known gD receptors, i.e. CD155, CD112. Such experiments, while of interest, are certainly not required to support the conclusions drawn here*.

We agree with the reviewer and plan to look at gD-specific antibodies and B cells in nectin and HVEM deficient mice. We have added this to the Discussion.

Reviewer #3:

*This manuscript describes a novel approach to HSV2 vaccination in mice. They find that deletion of the immunodominat gD protein in the virus elicits non-neutralizing antibodies which protect against challenge very impressively. They use a variety of readouts, all giving the same answer*.

*They show that the protection is mediated by antibody, because passive transfer protects though not if Ig depleted. They show that the protection is Fc dependent and go on to show that it is likely to act through ADCC. Whether this is the direct mechanism responsible is less clear but there is at least a good correlation between this assay in vitro and protection*.

*There are important general messages here. Neutralizing antibody is not essential for protection. That is known, though often ignored, in other virus infections; here it is very well illustrated and shown that such antibodies can be harnessed by vaccination to give protection. Secondly the immunodominant antigen may not elicit protective antibody but subdominant immunogens might; genetically manipulated vaccines offer a way to address that, though what determines immunodominance often remains a mystery*.

*Can the findings be generalized to other infections? They make a reference to the RV144 HV vaccine trial where some protection was given by non-neutralizing antibodies. It would be good to cite other more convincing examples, if they exist*.

We do not yet know if these findings can be generalized to other infections. There are few examples in the literature that demonstrate the role of non-neutralizing antibodies and most of the studies are in NHP models of SIV/SHIV and one study in influenza. We have added a few of these additional references as suggested.

*A small point that is relevant to human vaccinology. BALB/c and C57B6 mice are often used to test immunogens and for challenges. However in human terms these are just two individuals and may well be immunologically atypical compared to outbred mice or outbred humans. It would be good to test this protection in an outbred mouse model, if the same level of protection is seen that be very encouraging for a human vaccine. However this is not essential for acceptance of this manuscript*.

We agree with the reviewer and are planning future experiments looking at efficacy with outbred strains of mice (such as Swiss Webster).